# Soil and Plant Responses to Phosphorus Inputs from Different Phytase-Associated Animal Diets

Dario Fornara [1,*], Elizabeth M. E. Ball [1], Christina Mulvenna [1], Henry Reyer [2], Michael Oster [2], Klaus Wimmers [3], Hanne Damgaard Poulsen [4] and Arno Rosemarin [5]

1   Agri-Food & Biosciences Institute (AFBI), Newforge Lane, Belfast BT9 5PX, UK; Elizabeth.Ball@afbini.gov.uk (E.M.E.B.); Christina.Mulvenna@afbini.gov.uk (C.M.)
2   Research Institute for Farm Animal Biology (FBN), Wilhelm-Stahl-Allee 2, 18196 Dummerstorf, Germany; reyer@fbn-dummerstorf.de (H.R.); oster@fbn-dummerstorf.de (M.O.)
3   Faculty of Agricultural and Environmental Sciences, University Rostock, Justus-von-Liebig-Weg 7, 18059 Rostock, Germany; wimmers@fbn-dummerstorf.de
4   Department of Animal Science, Aarhus University, Blichers Allé 20, 8830 Tjele, Denmark; hdp@anis.au.dk
5   Stockholm Environment Institute, Linnégatan 87D, 10451 Stockholm, Sweden; arno.rosemarin@sei.org
*   Correspondence: dario.fornara@afbini.gov.uk

**Abstract:** The over-supplementation of animal feeds with phosphorus (P) within livestock-production systems leads to high rates of P excretion and thus to high P loads and losses, which negatively impact the natural environment. The addition of phytase to pig and poultry diets can contribute to reducing P excretion; however, cascading effects of phytase on plant–soil systems remain poorly understood. Here, we addressed how three different diets containing various levels of exogenous phytase, i.e., (1) no-phytase, (2) phytase (250 FTU), and (3) superdose phytase (500 FTU) for pigs (*Sus scrofa domesticus*) and broilers (*Gallus gallus domesticus*) might affect P dynamics in two different plant–soil systems including comfrey (*Symphytum × uplandicum*) and ryegrass (*Lolium perenne*). We found that differences in phytase supplementation significantly influenced total P content (%) of broiler litter and also pig slurry (although not significantly) as a result of dietary P content. P Use Efficiency (PUE) of comfrey and ryegrass plants was significantly higher under the intermediate 'phytase' dose (i.e., commercial dose of 250 FTU) when compared to 'no-phytase' and 'superdose phytase' associated with pig slurry additions. Soil P availability (i.e., water soluble P, WSP) in both comfrey and ryegrass mesocosms significantly decreased under the intermediate 'phytase' treatment following pig slurry additions. Dietary P content effects on P losses from soils (i.e., P leaching) were variable and driven by the type of organic amendment. Our study shows how commercial phytase levels together with higher dietary P contents in pig diets contributed to increase PUE and decrease WSP thus making the plant–soil system more P conservative (i.e., lower risks of P losses). Our evidence is that dietary effects on plant–soil P dynamics are driven by the availability of P forms (for plant uptake) in animal excretes and the type of organic amendment (pig vs. broiler) rather than plant species identity (comfrey vs. ryegrass).

**Keywords:** agricultural value chain; circularity; mixed crop-livestock system; phosphorus cycling





## 1. Introduction

Agricultural grasslands and croplands play a key role as livestock support systems and often receive large amounts of phosphorus (P) from animal excreta [1,2]. Increases in organic P additions to soils significantly enhance soil P availability and P surplus, which in turn is linked with greater P losses from runoff water or with additional P accumulation in soils [3,4]. Due to potential detrimental effects of P losses to the natural environment, there has been increasing urgency in improving P use efficiency (PUE) within intensive livestock-production systems [5,6]. One key cause of P surplus is the over-supplementation of animal

feeds, which is in turn associated with high P excretion rates and high P loads [7,8], which ultimately negatively impact soil and aquatic ecosystems [9].

A key challenge remains how to improve the sustainability of intensive livestock-production systems by closing the loop of the phosphate life cycle aiming to recover P from animal waste and utilise P in other sectors including agriculture [10]. A range of solutions have been proposed from both plant- and animal-centered perspectives. For example, new crop varieties with increased PUE could produce more biomass per gram of P absorbed or plants' capacity to extract immobile P from the soil could be enhanced by facilitating root-microbial symbioses [11]. In this context, varieties of comfrey (*Symphytum* spp.) are of interest for cultivation and usage in animal nutrition [12,13]. Comfrey contains relatively high levels of P, calcium, and potassium, as well as protein, which has the potential to establish local resource cycles and contribute to the remediation of over-fertilised soils as a crop for the future [12,13].

From an animal perspective, new feeding strategies need to be developed to improve P resource efficiency and reduce P losses to the environment [5]. Within intensive animal production systems, which include monogastric animals such as pigs and poultry, optimal P efficiencies require a combination of high P absorption, a sufficient animal skeletal storage, and a low P excretion [14]. Achieving this P balance in animal husbandry settings is challenging because most of the plant-derived P in cereal diets is bound up in the form of phytate P, which is associated with low digestibility [14]. This is primarily because the endogenous enzymatic capabilities of monogastric animals are limited to sufficiently cleave the phytate molecule and release P for absorption and utilisation. In comparison to phytate-bound P, inorganic phosphate is absorbed very effectively in pigs [15,16].

To increase P utilisation, the addition of various exogenous enzymes to pig and poultry diets has been evaluated [17]. Microbial phytase derived from the fungus *Aspergillus niger* catalyses the hydrolysis of phytate in the upper digestive tract, making orthophosphate available for uptake [18,19]. Previous experimental studies show that additions of phytase to animal diets contributed to reduce total P in animal wastes when compared to control diets without phytase [20,21]. It remains unclear, however, whether and how phytase-associated diets also affect P dynamics in soils or have any effect on PUE of cultivated plant species.

Evidence from incubation studies suggests that phytase addition to poultry diets can either decrease [22] or increase [23] P availability in soils (i.e., water soluble P, WSP). Other studies suggest that soil–plant available P (i.e., Olsen P) was reduced when phytase was added to pig diets [24] or that phytase addition to pig diets did not significantly affect P runoff from soils [25]. Such high variability in soil P dynamics could depend on both soil and manure-specific characteristics [26] but could also be due to lack of information on how plants may mediate soil responses to organic fertilisation. We are not aware of previous studies which have simultaneously addressed the response of specific plant species and soil P dynamics to the addition of animal wastes produced from different phytase diets.

To contribute to reducing this knowledge gap we specifically addressed how the supplement of three different exogenous phytase levels (i.e., no-phythase, phytase and superdose phytase) to the diet of two animal species, pig (*Sus scrofa domesticus*) and broiler (*Gallus gallus domesticus*) might potentially affect total P (%) content in slurries and litter, and how this in turn might be related to changes in P dynamics in two different plant–soil systems, including comfrey (*Symphytum × uplandicum*) and ryegrass (*Lolium perenne*).

We expect that the addition of phytase to animal diets will contribute to reduce total P in animal wastes when compared with control diets and in turn this will affect P dynamics in the comfrey- and ryegrass-soil system. These predictions were tested by measuring (1) total P (%) of animal wastes, (2) P Use Efficiency (PUE) of ryegrass and comfrey plants, (3) soil P availability (WSP), and (4) soil P losses (i.e., P in leachates) in a mesocosm experiment under controlled greenhouse conditions.

## 2. Methods and materials

### 2.1. Animal Trials

Experimental animal trials were carried out using Dan Duroc boars ($n = 80$ and $n = 32$) at the experimental farm of the Agri-Food and Biosciences Institute (AFBI) in Hillsborough (Northern Ireland) and male Ross 308 broilers ($n = 320$ and $n = 64$) at the AFBI Veterinary Sciences Division, Stormont, Belfast (Northern Ireland). To produce slurry and litter for this study, three experimental dietary treatments were applied to both pigs and broilers. Pig diet treatments were defined as: (1) 'no-phytase' = maize distiller dried grain solubles (DDGS) and rapeseed extract diet with no phytase formulated to contain 1.5% less energy, reduced amino acid content and P (i.e., 37% and 19% less P in grower and finisher formulations compared to treatment 2), (2) 'phytase' = a by-product based diet containing maize (DDGS) and rapeseed extract with commercial levels of phytase (250 FTU, 0.01%, Quantum® Blue, AB Vista, Marlborough, UK), formulated to contain recommended levels of energy, amino acids and P, and (3) 'superdose phytase' = the same formulation as the 'no-phytase' diet but including a phytase super dose (1000 FTU, 0.02%, Quantum® Blue, AB Vista, Marlborough, UK). Similarly, broiler diet treatments consisted of (1) 'no-phytase' = maize DDGS and rapeseed extract diet with no phytase formulated to contain less energy (i.e., 3.2% and 2.2% starter/grower and finisher diets, respectively) and reduced amino acid content and P (i.e., 30% and 32% less P in grower and finisher formulations, respectively), (2) 'phytase' = a maize DDGS and rapeseed extract diet with commercial levels of phytase (500 FTU, 0.01%, Quantum® Blue, AB Vista, Marlborough, UK), formulated to contain recommended levels of energy, amino acids and P and (3) 'superdose phytase' = the same formulation as the 'no-phytase' diet but including a phytase super dose (1500 FTU, 0.03%, Quantum® Blue, AB Vista, Marlborough, UK).

To produce pig slurry for this study, pig trials involved two three-week balance periods (11–14 weeks old and 14–17 weeks old) during which pigs were held in balance crates and offered one of the three diets. Separate samples of urine and faeces were collected daily for each pig and stored at 4 °C. Urine and faeces were mixed at the end of the collection periods to produce a representative sample of slurry over the finishing periods for use in the plant–soil mesocosm study. Broilers were placed in pens of 10 and offered production stage specific experimental diets from 0–42 days. Litter quantity was weighed at the end of the trial and litter was mixed from each pen according to treatment to obtain a representative sample for the mesocosm study.

### 2.2. Mesocosm Experiment

A mesocosm study was carried out under controlled conditions in a greenhouse setting located at AFBI headquarters in Belfast, UK. Root cuttings of Comfrey ("Bocking 14", Shipka, Bulgaria) were purchased from the Balkan Ecology Project (www.balkep.org; access date 5 June 2018) in June 2018. Seeds of Ryegrass (*Lolium perenne*, variety: Fintona) were obtained from the AFBI grass breeding programme at Loughgall, Northern Ireland, UK. Root cuttings (one per pot) and grass seeds (0.33 g) were added to pots (10-L capacity), which contained an air-dried mix of sterilised soil and sand (70% soil:30% sand mix ratio). Water Holding Capacity (WHC) of the soil/sand mix was determined and water was added to achieve 50% of soil WHC. Inorganic fertilisers (NPK 16-10-24; Solabiol, Ecully, France) were added to establish plants until the end of September 2018 while pots were continually held at 50% WHC. Aboveground plant biomass was then removed and belowground plant compartments remained dormant over winter. In March 2019, comfrey and ryegrass plants started re-growing and pots were arranged in a randomised block design which included 2 plant species (ryegrass and comfrey) × 2 animal species (pig and broiler) × 3 organic amendment treatments from the three different animal diets ('no-phytase', 'phytase' and 'superdose phytase') × 3 blocks + 2 control pots (within each block) = 42 mesocosms. Pig slurry and broiler litter were first dried on a steaming water bath to drive off volatile moisture before being placed in an oven at 105 °C for 16 h. The oven dry materials were cooled in a desiccator and then milled using a Culatti hammer mill (Glen Creston

Ltd., London, UK) in preparation for total Nitrogen (%N) measurements using a LECO Trumac CN Analyser (St. Joseph, MI, USA). Total P in dry material was analysed using a two-step protocol; first, organic phosphorus and polyphosphates were converted to ortho-phosphate (ascorbic acid method), and secondly, ortho-phosphate was determined based on a colorimetric method [27]. Total P and N content were based on dry weights (DW) of pig slurry (DW = 9.5%) and broiler litter (DW = 37.4%). Between 16 April and 24 May 2019, two applications of organic amendments were carried out to deliver fertilisation rates of 30 kg P ha$^{-1}$ and 250 kg N ha$^{-1}$. Extra water was added to pots receiving broiler litter to make sure WHC of each pot was maintained at 50%. Control pots only received one dose of triple super P (inorganic fertiliser) initially to support seedling growth under same rates of P and N fertilisation.

### 2.3. Aboveground Plant Parameters

On the 9th of May and 12th of June 2019, 1000 mL of water were added to each pot and after 30 min the soil solution leachate was collected from the pot saucers, weighed and sent to the laboratory for WSP analyses. The pots were then retained at 50% WHC. Ryegrass growth rate was higher compared to comfrey and we thus harvested ryegrass biomass on 5th of June 2019; mass was weighed, dried and retained to calculate overall productivity at the end of the mesocosm experiment. The pots were retained at approx. 50% water holding capacity. Weights and amount of water added were recorded in a laboratory notebook. On 3 July 2019 all plant biomass from comfrey and ryegrass pots was harvested, weighed and dried. After weights were recorded all plant material was milled and sent for chemical analyses. Total P in plant material was analysed as above following the colorimetric method [27].

### 2.4. Soil and Root Parameters

On the 5 July 2019, 5 cm diameter soil cores were taken from each individual pot to be analysed for WSP. Water-extractable P provides a good indication of plant-available P [28]. On the 10 July 2019, three 20 cm deep soil cores were taken from each pot to estimate root mass. Soil cores were washed over a 2 mm mesh size, roots collected and dried at 70 °C for three days. Dried roots were weighed and then ground through a 1 mm hammer mill and sent for analysis. Total P in root material was analysed in the same way as for aboveground plant biomass.

### 2.5. Microbiota Profiling

Representative soil cores measuring 50 mm × 10 mm (~8 g) were collected from each pot before the growing season (March 2019) and at harvest (July 2019). In detail, samples were collected before and after application of the organic amendments (pig manure, chicken litter) generated by varying amounts of phytase feed additives (no-phytase, phytase, superdose phytase) for both comfrey and ryegrass species. In addition, samples from control pots of each plant species were taken at both time points. This resulted in a total of *n* = 42 samples per time point, with three replicates pooled at each time point, providing 28 samples for downstream analyses. Microbial DNA from the soil samples was extracted with the DNeasy PowerLyzer PowerSoil Kit (QIAGEN, Hilden, Germany) as previously described [29]. PCR amplification of the 16S rRNA gene was performed in duplicate for each sample using primers targeting the V4 variable region (515′F and 806R) [30]. The KAPA HIFI HotStart kit (Roche Diagnostics, Mannheim, Germany) was used for PCR under the following cycling conditions: 95 °C for 2 min, 30 cycles of 95 °C for 30 s, 55 °C for 60 s and 72 °C for 90 s and terminal 72 °C for 10 min. All amplicons were combined in equal amounts and sequenced on a HiSeq2500 instrument (Illumina, San Diego, CA, USA).

### 2.6. Data Analysis

Potential effects of plant and animal species identity and type of organic amendment (produced by the three different animal diets) on multiple plant and soil parameters were

tested using ANOVA. Significant differences between factor levels (i.e., phytase-associated diets) were assessed using a post hoc Tukey's test. We also calculated Phosphorus Use Efficiency (PUE) to assess plants' ability to take up P under different nutrient treatments. PUE was calculated as total plant P uptake (i.e., grams of P in dry plant mass) divided by total P inputs (i.e., grams of P added to the soil). All analyses were performed using JMP version 9.0.0 (SAS Institute, Cary, NC, USA, 2010).

The 16S rRNA sequencing data was processed with the mothur software (version 1.44.1) [31]. For each sample, at least 74,600 sequencing reads were considered. For the alignment of the sequence data and the taxonomic assignment of the operational taxonomic units (OTU), the Silva reference database (version 138) was used. Information from the sample-specific microbial community data were condensed using a non-metric multidimensional scaling (NMDS) approach, and the distances between samples were visualised in an NMDS plot using the vegan package (version 2.5-7) implemented in R software (version 4.1.2; R core team, Vienna, Austria). Statistical significance between the microbial communities of the respective groups (2 plant species, 3 organic amendment treatments and 3 different animal feeds) was tested using the analysis of similarity (ANOSIM) approach (vegan, v2.5-7). Moreover, the microbial phyla mainly contributing to the divergences between the different groups were derived by sparse Partial Least Squares Discriminant Analysis (sPLS-DA) using the R package "mixOmics" (version 6.10.6) [32]. The variable selection process considered 3 microbial factors for each of the first two components to analyse the effects of (i) plant species, (ii) organic amendments by animal species and (iii) phytase feed additives.

## 3. Results

### 3.1. Effects of Animal Diet on the P Content of Animal Manures

Table 1 shows the N and P content of pig slurry and broiler litter on dry matter basis (DMB). We found that the P content of pig slurry from the "phytase" treatment ($1.76 \pm 0.31\%$ DM) was higher but not significantly different from the P content of the slurry from the "no-phytase" ($1.42 \pm 0.22\%$ DM) and "super phytase" ($1.34 \pm 0.21\%$ DM) treatments. The P content of broiler litter from the "phytase" treatment ($0.96 \pm 0.06\%$ DM) was significantly ($p < 0.01$) higher than the P content of the litter from the "no-phytase" ($0.84 \pm 0.10\%$ DM) and "super phytase" ($0.85 \pm 0.07\%$ DM) treatments. There was no consistent treatment effect across pig slurry and broiler litter in relation to N content.

**Table 1.** Total phosphorus and nitrogen content (%) of pig slurry and broiler litter produced by animals under different diets characterised by varying phytase levels. 'no-phytase', 'phytase' (commercial levels at 250 FTU or 500 FTU Quantum® Blue, for pigs and broilers, respectively), and 'superdose phytase' (1000 FTU or 1500 FTU Quantum® Blue, for pigs and broilers, respectively). No significant differences were detected among %P and %N content across the different diets except for P% content of broiler litter, which was higher under the 'Phytase' treatment.

|  | Phosphorus Content | | Nitrogen Content | |
|---|---|---|---|---|
|  | Pig Slurry (% DM) | Broiler Litter (% DM) | Pig Slurry (% DM) | Broiler Litter (% DM) |
| By-product diet + superdose phytase | $1.34 \pm 0.21$ | $0.85 \pm 0.07$ | $8.89 \pm 0.31$ | $5.03 \pm 0.48$ |
| By-product diet + phytase | $1.76 \pm 0.31$ | $0.96 \pm 0.06$ * | $8.50 \pm 0.31$ | $4.89 \pm 0.34$ |
| By-product diet no-phytase | $1.42 \pm 0.22$ | $0.84 \pm 0.10$ | $8.95 \pm 0.22$ | $4.60 \pm 0.35$ |

* $p < 0.05$.

### 3.2. Effects of Animal Organic Amendments on Plant Biomass and P Mass

Compared to control pots we found that organic amendments from either pigs or broilers significantly increased plant growth in terms of total plant biomass and P mass (i.e., grams of plant P) in shoots and roots (Table 2; Figure 1). In particular, we found that shoot mass and P mass in shoots increased significantly compared to the control (Table 2; $p < 0.0001$ for both analyses). Root mass and P mass in roots were generally higher under organic amendments but not significantly different from control root mass (Table 2, $p > 0.05$). Plant and P mass (roots and shoots) was significantly higher ($p < 0.001$ for all analyses) in comfrey than ryegrass regardless of organic nutrient treatment (Table 2). Pig slurry additions significantly increased plant aboveground and belowground biomass (for both comfrey and ryegrass; $p < 0.0001$) more than broiler litter additions (Figure 1a,b).

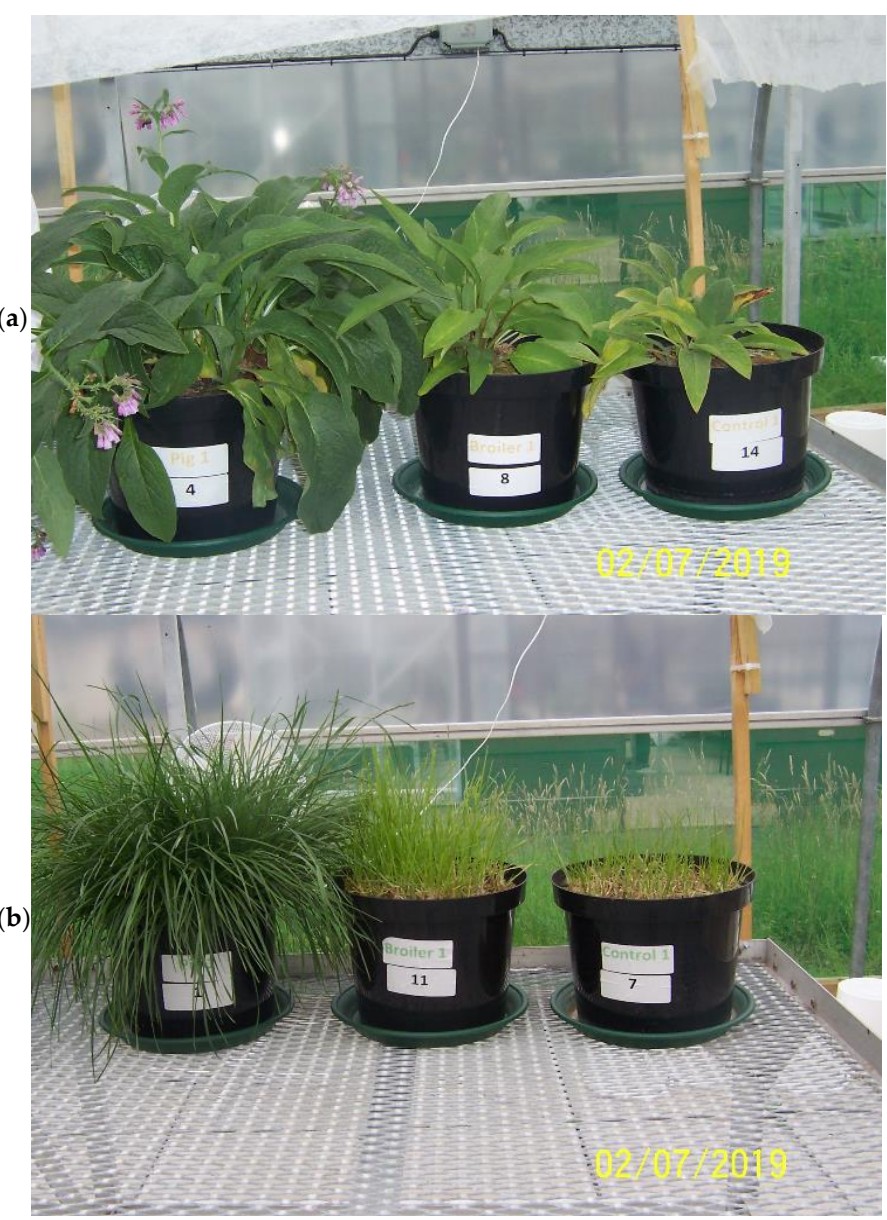

**Figure 1.** Example of growth response of two plant species namely, Comfrey (*Symphytum × uplandicum*; (**a**)) and Italian ryegrass (*Lolium multiflorum*; (**b**)) to different fertilisation treatments including (from **right to left** in the photo) control (no fertiliser applied), broiler litter and pig slurry all applied at rates of 30 kg P ha$^{-1}$ and 250 kg N ha$^{-1}$. The photo was taken on 2 July 2019 after a ~3 month-long growing period under controlled greenhouse conditions.

**Table 2.** Effects of two different organic amendments (pig slurry and broiler litter) from different animal diets on plant aboveground biomass (shoot biomass, grams), shoot P mass (grams of P), plant belowground biomass (root biomass, grams) and root P mass (grams of P) of two different plant species comfrey (*Symphytum × uplandicum*) and Italian ryegrass (*Lolium multiflorum*). Animal diets include: control = only inorganic fertiliser applied once (i.e., one dose of triple super P initially applied to support seedling growth), no-phytase, phytase and superdose phytase. Different letters indicate significant differences among treatments (Tukey HSD test).

| | Pig Slurry | | | | | | | | Broiler Litter | | | | | | | |
| | Comfrey | | | | Ryegrass | | | | Comfrey | | | | Ryegrass | | | |
| | Control | No Phytase | Phytase | Super-phytase | Control | No Phytase | Phytase | Super-phytase | Control | No Phytase | Phytase | Super-phytase | Control | No Phytase | Phytase | Super-Phytase |
|---|---|---|---|---|---|---|---|---|---|---|---|---|---|---|---|---|
| Shoot biomass (g) | 10.2 ± 5.63 c | 74.2 ± 4.35 a | 77.4 ± 5.76 a | 64.5 ± 5.16 b | 1.93 ± 1.25 b | 55 ± 4.32 a | 54.8 ± 4.66 a | 54.5 ± 5.13 a | 10.2 ± 5.6 b | 21 ± 2.3 a | 22.3 ± 2.7 a | 21.5 ± 5.1 a | 1.93 ± 1.2 b | 4.5 ± 0.43 a | 4.7 ± 0.46 a | 4.6 ± 0.51 a |
| Shoot P mass (g) | 0.02 ± 0.01 c | 0.8 ± 0.01 b | 0.14 ± 0.01 a | 0.8 ± 0.01 b | 0.02 ± 0.01 | 0.1 ± 0.01 | 0.11 ± 0.01 | 0.2 ± 0.01 | 0.02 ± 0.01 | 0.05 ± 0.01 | 0.03 ± 0.01 | 0.05 ± 0.01 | 0.02 ± 0.01 | 0.007 ± 0.001 | 0.008 ± 0.001 | 0.008 ± 0.001 |
| Root biomass (g) | 2.2 ± 1.62 b | 9.73 ± 2.3 a | 6.1 ± 1.7 a | 5.75 ± 5.1 ab | 0.7 ± 0.3 c | 2.85 ± 0.8 a | 1.32 ± 0.6 b | 1.35 ± 0.7 b | 2.2 ± 1.6 b | 6.93 ± 3.3 a | 7.46 ± 2.7 a | 10 ± 3.1 a | 0.7 ± 0.3 b | 1.29 ± 0.23 a | 1.33 ± 0.3 a | 1.08 ± 0.25 a |
| Root P mass (g) | 0.3 ± 0.49 | 2.7 ± 0.5 | 1.33 ± 0.7 | 1.27 ± 0.4 | 0.08 ± 0.01 b | 0.47 ± 0.5 a | 0.23 ± 0.7 a | 0.34 ± 0.4 a | 0.3 ± 0.49 b | 1.57 ± 0.56 a | 1.18 ± 0.73 a | 2.1 ± 0.68 a | 0.08 ± 0.01 b | 0.13 ± 0.02 a | 0.14 ± 0.03 a | 0.12 ± 0.03 a |

### 3.3. Effects of Animal Diet on Plant Phosphorus Use Efficiency (PUE)

We found that significant changes in plant Phosphorus Use Efficiency (PUE) were driven by animal species identity whereby broiler litter was associated with higher PUE than pig slurry regardless of diet treatment ($p < 0.0001$; Figure 2a,b). Pig slurry applications associated with the commercial 'phytase' treatment (250 FTU, 0.01%, Quantum® Blue) significantly increased PUE in both ryegrass and comfrey plants when compared to 'no-phytase' or 'superdose phytase' treatments ($p = 0.007$; Figure 2a). Similarly, broiler litter applications associated with the commercial 'phytase' treatment increased PUE in comfrey plants (Figure 2b; $p = 0.002$) but not in ryegrass plants (Figure 2b). Control plants (not fertilised) showed higher PUE but only for ryegrass either under pig slurry or broiler litter applications, whereas control comfrey plants did not show any significant PUE increase.

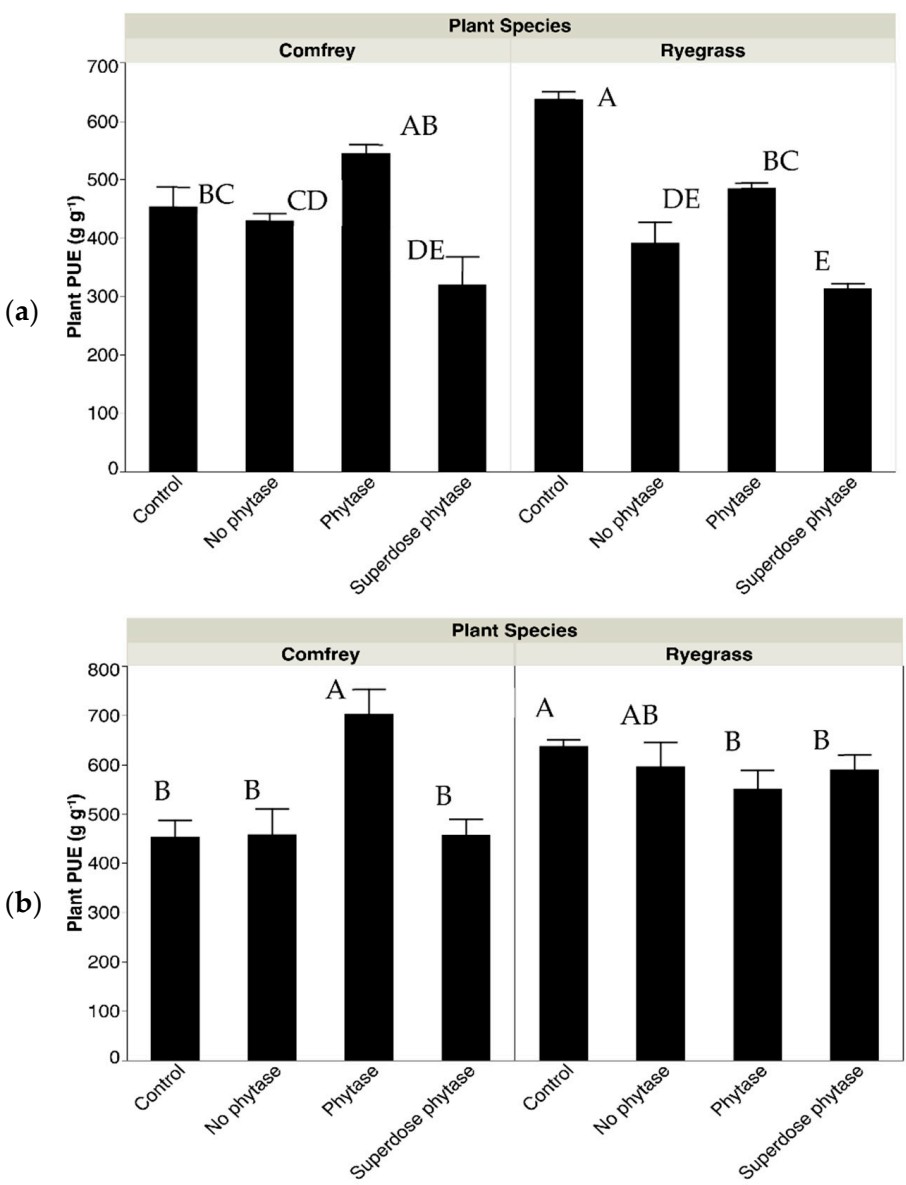

**Figure 2.** Effects of two different organic amendments (**a**) pig slurry and (**b**) broiler litter from different animal diets on plant phosphorus use efficiency (PUE) of two different plant species comfrey (*Symphytum* × *uplandicum*) and Italian ryegrass (*Lolium multiflorum*). Animal diets include: control = only inorganic fertiliser applied once (i.e., one dose of triple super P initially applied to support seedling growth), no-phytase, phytase and superdose phytase. Different letters indicate significant differences among treatments (Tukey HSD test).

### 3.4. Effects of Animal Diet on Soil P Availability

We found that, under pig slurry applications, soil P availability (i.e., WSP) significantly decreased under the 'phytase' treatment when compared to 'no-phytase' and 'superdose phytase' ($p$ = 0.02; Figure 3a). However, broiler litter applications did not have any significant effect on soil P availability under comfrey plants, but increased soil P availability associated with 'phytase' and 'superdose phytase' under ryegrass plants (Figure 3b).

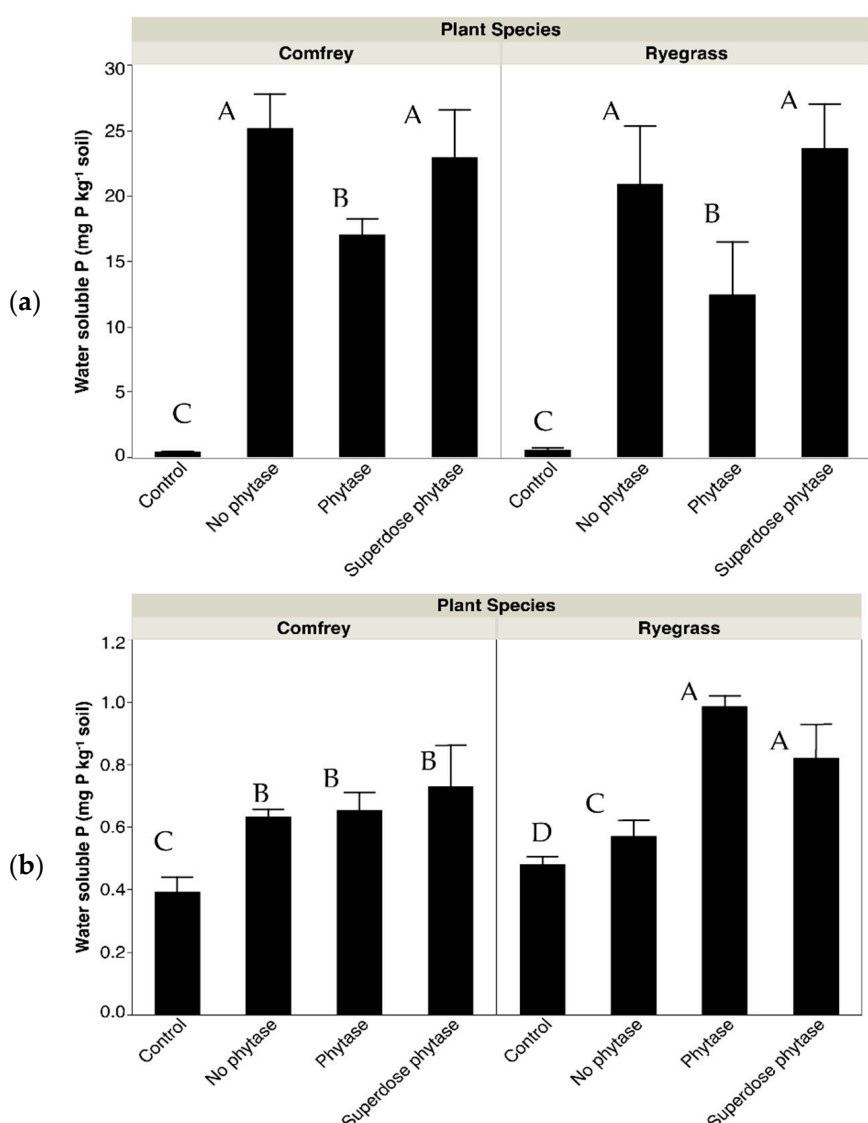

**Figure 3.** Effects of two different organic amendments (**a**) pig slurry and (**b**) broiler litter from different animal diets on soil available P (i.e., WSP) of two different plant species comfrey (*Symphytum × uplandicum*) and Italian ryegrass (*Lolium multiflorum*). Animal diets include: control = only inorganic fertiliser applied once (i.e., one dose of triple super P initially applied to support seedling growth), no-phytase, phytase and superdose phytase. Different letters indicate significant differences among treatments (Tukey HSD test).

### 3.5. Effects of Animal Diet on P Loss in Leachates

We found that soil P loss in leachates was lowest in control pots when compared to pig slurry treatments (Figure 4a) but not significantly different from soil P loss in broiler litter treatments (Figure 4b). In general, soil P levels in leachates were higher under pig slurry additions compared to broiler litter (although not statistically different) and at least under pig slurry applications, P losses were higher under the 'no-phytase' treatment.

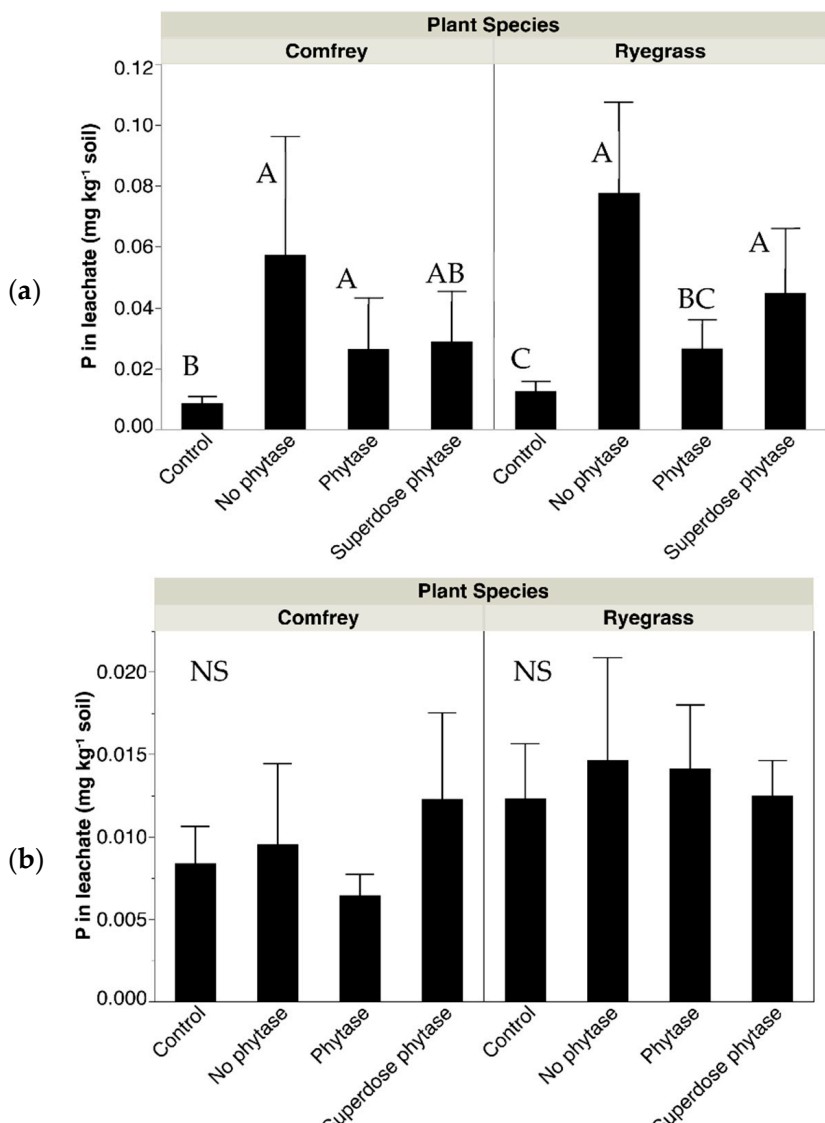

**Figure 4.** Effects of two different organic amendments (**a**) pig slurry and (**b**) broiler litter from different animal diets on P loss (i.e., WSP) in leachates from experimental pots planted with two different plant species, comfrey (*Symphytum* × *uplandicum*) and Italian ryegrass (*Lolium multiflorum*). Animal diets include: control = only inorganic fertiliser applied once (i.e., one dose of triple super P initially applied to support seedling growth), no-phytase, phytase and superdose phytase. Different letters indicate significant differences among treatments (Tukey HSD test).

*3.6. Effects of Plant Species, Organic Amendment, and Animal Feed on Soil Microbiota Profiles*

The analyses of microbiota composition revealed pronounced effects of organic amendment and plant species (Figure 5). The NMDS analysis maps two dimensions, with the first level mainly differentiating between inorganically fertilised soil as well as pig slurry and broiler litter (NMDS1). In addition, the second dimension showed an effect of plant species, with soil microbiota samples differing between comfrey and ryegrass pots (NMDS2). Indeed, the analysis of similarity (ANOSIM) revealed significant differences in microbial composition between organic amendments (R: 0.60; $p < 0.001$) and plant species (R: 0.26; $p < 0.001$). With regard to the effects of plant species on the composition of the soil microbiota at phylum level, the sPLS-DA analyses imply that *Fibrobacterota* and *Nitrospirota* (more abundant in ryegrass samples) and *Hydrogenedentes* and *Gemmatimonadota* (more abundant in comfrey samples) provide the highest contributions to discrimination (Supplemental Figure S1A). Moreover, the sPLS-DA components revealed the abundance

of *Actinobacteriota*, *Proteobacteria*, *Acidobacteriota* and *Euryarchaeota* as major discriminating variables between organic pig amendment, broiler litter amendment and unfertilised soil (Supplemental Figure S1B). Microbial components identified for the discrimination of the different animal feeds comprised *Fusobacteriota*, *Zixibacteria* and *Hadarchaeota* (Supplemental Figure S1C).

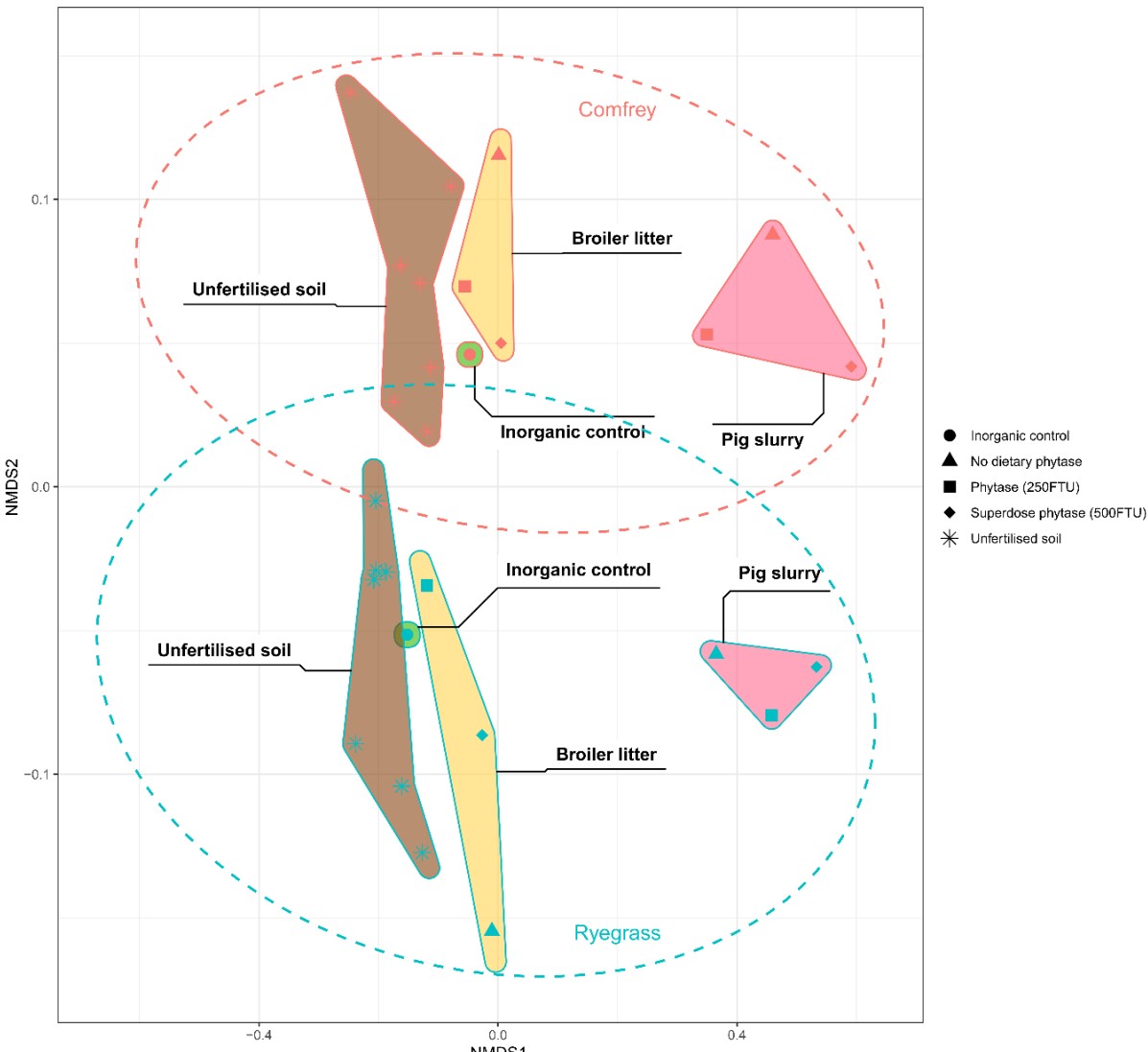

**Figure 5.** Non-metric multidimensional scaling (NMDS) of the soil microbiota composition based on 16S rRNA sequencing data. The category "unfertilised soil" comprises samples taken at the beginning of the experiment, i.e., before the fertiliser application. Alteration of the microbiota composition was pronounced due to application of organic amendments (pig slu, chicken litter) generated by varying amounts of phytase feed additives (no phytase, phytase, superdose-phytase) in comfrey and ryegrass specimen.

## 4. Discussion

### 4.1. Dietary Phytase Effects on P Excretion

Our findings show that the P content of pig slurry and broiler litter tended to be higher under the "phytase" treatment (i.e., commercial phytase levels at 250 FTU or 500 FTU Quantum® Blue, for pigs and broilers, respectively), when compared to the 'no-phytase' or 'superdose phytase' treatments. This is possibly because the 'phytase' treatment was associated with higher dietary P contents, thus suggesting that high P levels in slurry and litter may partly reflect the P content of the diet. In fact, dietary P diets may prompt animal-

intrinsic endocrine responses with respect to renal mineral excretion rates [29]. Previous studies have shown that when overall total dietary P is reduced, total P excretion after the supplementation of phytase to pig [33] or broiler diets [34] is also reduced. However, animal genetics can also influence P utilisation whereby about 40% of the phenotypic variation in blood P can be explained by genetic variation [35]. A key point to clarify in further studies is how the interaction between animal performance and phytase diet may influence P excretion [36,37] and thus the amount of P that is released into the environment.

Differences in total P excretion in our study could have different explanations. First, while the "no-phytase" treatment contained no supplemental exogenous phytase, endogenous phytase might not be deactivated during the cold pelleting process of diet production and could have thus affected total P of animal wastes. Second, variable phytase supplementation effects on P excretion could be related to differences in experimental methodologies and the efficacy of phytase and other factors influencing phytase activities [17] including the effect of different forms of exogenous microbial phytase [38]. For example, a previous study addressing the effects of phytase supplementation on broiler diets shows that broilers fed with wheat and triticale had significantly higher (and not lower) total P excretion than those fed maize-based diets, which in turn had significantly higher P excretion than broilers fed barley-based diets [39]. Third, it is also possible that measuring total P in animal excreta does not accurately capture potential phytase dietary effects on P dynamics. This could be the case in poultry diets, which mainly consists of cereals where P is stored in an organic form (i.e., phytic acid), and the addition of phytase tends to decrease this organic P share while increasing the share of inorganic P (i.e., available reactive P) [40]. It has been shown in previous studies that the supplementation of phytase to broiler diets contributed to increase P availability in the animal manures [34,39,41]. There is also evidence, however, that phytase supplementation to broiler diets did not have any effect on P availability in the litter [42–44].

Changes in P availability in response to phytase diets are, however, different in pigs, where it is has been shown that water-soluble P in excretes is reduced when phytase is added to diets [21] or is not affected by phytase supplementation [45]. More studies are needed to address the underlying physiological mechanisms and processes responsible for such high variability associated with phytase effects on total and available P in animal wastes.

### 4.2. Dietary Phytase Effects on the Plant–Soil System

We found significant effects of phytase diets on the plant–soil system that we set up under controlled greenhouse conditions. In general, pig slurry additions significantly increased both plant aboveground and belowground biomass (either in ryegrass or comfrey) more than broiler litter additions (Figure 1a,b). This agrees with the results of a pot experiment where barley plants shown greater grain yields when received pig slurry than broiler litter additions [40]. Higher plant yields in our study under pig slurry applications can be partly explained by the fact that P availability in soils (i.e., WSP) was >20 times higher in pig slurry-amended soils than in soils receiving broiler litter (Figure 3a vs. Figure 3b). Interestingly, organic amendments obtained from the pig and broiler trials differed considerably with respect to the composition of the microbiota (Figure 5). It has been shown recently that the application of organic fertiliser such as cattle slurry can shape the soil microbiota community, possibly because of changes in the supply of micro- and macronutrients to soils, but also changes in soil physical and chemical properties [46,47]. It has also been shown that the composition of the microbiota from the mammalian gut and the soil/root system is fundamentally different and thus amendment-born microbes might not necessarily persist in soil ecosystems [48]. Further studies should address how differences in microbial taxa associated with different plant species and/or organic amendments translate into different microbial activities that influence P dynamics in soils.

In terms of phytase diet treatments, we found that under pig slurry additions P Use Efficiency (PUE) of comfrey and ryegrass plants was significantly higher, while WSP in



soils was significantly lower at the intermediate commercial 'phytase' (250 FTU) supplementation level when compared to 'no-phytase' and 'superdose phytase' (1000 FTU) treatments. These results suggest a non-linear response of plant PUE (Figure 2a) and soil P availability (Figure 3a) to increasing levels of phytase in pig diets, which could result from the complex interaction between P availability in animal excretes, soil P dynamics, and plant P requirements. There have not been studies so far which have addressed whether increasing phytase levels (e.g., 0 to 1000 FTU) in animal diets may have cascading effects on soils and plants. Our results partly agree with those of a recent soil incubation study, which show that 'no-phytase' treatment was associated with higher soil–plant P availability (measured as Olsen P index) than a dietary treatment of 500 phytase units (FTU) [24]. As both plant species (comfrey and ryegrass) showed similar responses (high PUE, low WSP) to intermediate 'phytase' treatments, it is likely that such responses were driven by phytase and dietary P content effects on the quality and availability of P forms in pig excretes. Further studies need to address how varying levels of phytase supplementation and dietary P (and N) content could interact and influence excretion of WSP [21,49,50] and how this in turn may affect WSP in soils.

In contrast, plant responses to broiler litter additions were more variable and did not show predictable changes in PUE and WSP in comfrey or ryegrass mesocosms. It could be that overall lower soil P availability (lower WSP) in soils fertilised with broiler litter has also affected plant PUE by limiting the amount of P available to plant uptake. Lower WSP in broiler litter amended soils could be partly due to experimental setting limitations and in particular to the length of the mesocosm trials, which only run for 3 months and thus not enough to incorporate P inputs from broiler litter (which is more dense than pig slurries), into the soil system.

We found that phytase-associated dietary effects on P losses from soils (i.e., P leaching) were variable and driven by the type of organic amendment. In particular, pig slurry additions were associated with higher P losses than broiler litter possibly because of the liquid nature of pig waste amendments. Although not statistically significant, we observed that P leaching was lower under the 'phytase' treatment, which would agree with the high PUE and low WSP results that we found under pig slurry additions. This suggests that intermediate 'phytase' supplementation (250 FTU) may be associated with a more conservative use of P within the soil–plant system when compared with 'no-phytase' or 'superdose phytase' treatments. Previous studies show that phytase supplementation to broiler diets had no significant effect on P leaching from soils [26,34]. However, it has been shown that wet storage of broiler litter contributed to increase concentrations of water-soluble P, which then increased reactive P in runoff from litter-amended soils [41].

Overall, our findings suggest that the supplementation of commercial phytase levels (i.e., 250 FTU) together with higher dietary P contents can result in higher P content of animal wastes. Instead, superdose phytase additions to animal diets associated with lower dietary P contents resulted in lower P excretion.

Our evidence is that commercial phytase levels together with higher dietary P contents added to pig diets contributed to increase PUE and decrease WSP, thus making the plant–soil system more P conservative. Thus, despite commercial phytase levels increased P excretion especially when associated with high dietary P levels, the plant–soil system was more efficient in acquiring and retain the excreted P. Our study suggests that phytase dietary effects on plant–soil P dynamics are driven by the availability of P forms (for plant uptake) generated by the animal diet and the type of organic amendment (pig vs. broiler) rather than plant species identity (comfrey vs. ryegrass). The results show that insights into soil–plant–animal interactions are crucial to optimise P fluxes along the value chain and to minimise losses to the environment and we suggest that further studies should address these interactions making use of plot-scale field experiments.

**Supplementary Materials:** The following supporting information can be downloaded at: https://www.mdpi.com/article/10.3390/agronomy12010130/s1. Supplementary Figure S1. Discriminant analyses of soil microbial profiles in the comparison of plant species (A), organic amendment treatments (B) and animal feeds (C). The plots represent the distances between the samples and the separation of the groups with the ellipse indicating the 95% confidence interval. The contributions of the most important phyla to each of the two components shown are represented with the colour identifying the group with the highest mean abundance of the respective variable.

**Author Contributions:** Conceptualization, D.F.; Methodology, D.F., E.M.E.B. and C.M.; software, H.R. and M.O.; writing—original draft preparation, D.F.; writing—review and editing, all Authors. All authors have read and agreed to the published version of the manuscript.

**Funding:** This work has received funding from the European Research Area Network (ERA-NET) co-funds on Sustainable Animal Production (SusAn) as part of the PEGaSus project (2817ERA02D).

**Institutional Review Board Statement:** Not applicable.

**Informed Consent Statement:** Not applicable.

**Data Availability Statement:** The Agri-Food Biosciences Institute (AFBI) owns the data presented in this study under the database rights. AFBI requires that specific agreements are put in place with regard to confidentiality before data can be disclosed to third parties.

**Acknowledgments:** This research has received funding from the European Research Area Network (ERA-NET) co-funds on Sustainable Animal Production (SusAn) as part of the PEGaSus project (2817ERA02D) and the Department for Environment, Food and Rural Affairs (DEFRA) in the UK. The co-authors do not have any conflict of interest to declare. We would like to thank Elizabeth-Anne Wasson, Elizabeth Mulligan, Natasha Crumlish, Joseph Larkin, Gareth Ridgway, Brian Wallace and Hugh McKeating for their assistance with greenhouse work and laboratory analyses. The authors thank Angela Garve and Aisanjiang Wubuli for excellent technical help in microbiota analyses.

**Conflicts of Interest:** The authors declare no conflict of interest.

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
