# Peer review of "Soil and Plant Responses to Phosphorus Inputs from Different Phytase-Associated Animal Diets"

_agronomy, doi:10.3390/agronomy12010130_

Round 1

Reviewer 1 Report

The manuscript is focused on an important issue of reducing P losses  to the environment by the addition of phytase to pig and poultry diets. In general it is well written, nevertheless, it requires minor adjustments.

  1. In the Abstract the sentence “We found that differences in phytase supplementation affected total P content (%) of pig slurry and broiler litter as a result of dietary P content” is not entirely consistent with interpretation in lines 224-226.
  2. Please, standardize the spelling of phytase: no-phytase, No-phytase, superdose phytase, Superdose Phytase etc. in the manuscript.
  3. Please, explain carefully wat was the control in the experiment: in the material part, there is the information “ inorganic fertilizers (NPK 16-10-24) were added to establish plants until the end of September” and in Table 2 “Control=no fertilizer applied”, and then on page 10, “We found that soil P loss in leachates was lowest in control pots
    receiving pig slurries (Fig. 4a)”
  4. Line 152: %N
  5. Line 231: Table 1. Total phosphorus
  6. Line 234: %P and %N
  7. What doses of mineral NPK were applied in the experiment?
  8. Please, add the letters to Fig. 4 b

Reviewer 2 Report

Reviewer comments

Agronomy (ISSN 2073-4395)

Manuscript ID: agronomy-1519779

Manuscript Title: Soil and plant responses to phosphorus inputs from different phytase-associated animal diets"

In this research, the authors wrote an article about “Soil and plant responses to phosphorus inputs from different phytase-associated animal diets” The topic is nice and fits well with the scope of Agronomy-MDPI, and this point is of interest for the scientific community. The table, figures, and graphics are prepared in good quality and well structured. However, the text needs a major revision before publication.

The comments:

Abstract:

  • Numerical data or ratios should be put in place to show the effects of Nano-materials treatments that have given outstanding results even from the previous results.
  • Please give included the impact of the different kinds of Nano-materials especially the modern ones on plant growth and productivity in the abstract section.
  • Lines 28-29: “Soil P availability (i.e. water soluble P, WSP) in both comfrey and ryegrass mesocosms significantly decreased under the intermediate ‘Phytase’ treatment following pig slurry additions”. Pls explain and add specific information in this regard.
  • It is preferable if you can add graphical abstract so that it is easier to understand and be clearer.

Keywords: I suggest rephrasing some words because keywords should not repeat words from the title.

Introduction:

  • The usage of abbreviation should be used after the full term. Please be consistent with the usage of all abbreviations. Pls revise the abbreviations in the whole parts of this review article.
  • The current state of this review article needs some more attention.
  • Lines 52-55: “A range of solutions have been proposed from both plant- and animal-centered perspectives. For example, new crop varieties with increased PUE could produce more biomass per gram of P absorbed, or plants’ capacity to extract immobile P from the soil could be enhanced by facilitating root-microbial symbioses [11]. ”. Please shorten these sentences”.
  • Lines 57-59: “Comfrey contains relatively high levels of P, calcium, and potassium, as well as protein, which has the potential to establish local resource cycles and contribute to the remediation of over-fertilized soils as a crop for the future” pls add relevant ref.
  • Lines 64-66: “Achieving this P balance in animal husbandry settings is challenging because most of plant-derived P in cereal diets is bound up in the form of phytate P, which is associated with low digestibility. pls add relevant ref.

Methods and materials

  • The materials and methods part is clear and well written.
  • Lines 162: “Soil and plant parameters” pls divide this part into two divisions”, the first one is soil parameters and the second is plant parameters. This is the first. The second is, Please explain and write these parameters in more detail because it is a very important part of the research.
  • General comment in the materials and methods part; Pls, shorten the long sentences as soon as possible and avoid repetition.

Results:

  • The figures should be reproduced well because their current form is not good and it is not appropriate to be placed in the article in this form. This is open access Journal, so you can exchange these figures with colored ones and more clear.
  • Lines 249-251:” Table 2. Effects of two different organic amendments (pig slurry and broiler litter) from different animal diets on plant aboveground biomass (shoot biomass, grams), shoot P mass (grams of P), plant belowground biomass (root biomass, grams), and root P mass (grams of P) of two different plant species comfrey (Symphytum x uplandicum) and Italian ryegrass (Lolium multiflorum). Animal diets include: Control = no fertilizer applied, No Phytase, Phytase, and Superdose Phytase”. In this table, it is not clear that there is any evidence of the existence of significant differences between the treatments, are there no significant differences between the treatments, or did the authors not include anything that shows that.
  • Please, make an effort to synthesize the text avoiding redundancies and repetitions in this part.
  • The results part is too long. Please shorten this section as much as possible, taking into account the lack of violation of the content and the non-repetition.
  • This part should be better organized and extended. It is important to try to better deepen and explain.

Discussion

  • The discussion part is short relative to the big data in the results part. Please add more discussion for more explanation especially in the important parameters as much as possible, taking into account the lack of violation of the content and the non-repetition.
  • Some discussion sentences need clarification and interpretation, and recent references need to be used as much as possible.
  • The discussion part should be better organized and extended. It is important to try to better deepen and explain.

Conclusions; Do not repeat the above sentences in the conclusion part. In conclusion, you should write a summary of your work in short sentences so that I, as a reader of this article, can understand what the article ended up being.

  • There is another important part that must be added at the end of this article, which is the current situation of using Phytase and Super-phytase on the plant growth and production and adding the vision or future expectations of this system of nutrition in order to increase the productivity of crops as well as from an economic point of view and in terms of its impact on protecting the environment from pollution, ...etc.

References:

  • The number of references is 50 ref. (only 6 references in the last five years) I think it is not enough. So, pls delete the old ones and add the recent references (only from 2019 to 2022) with avoiding repetition and self-citation as soon as possible.
  • There are several recent ref. in the same trend of the topic of this MS, pls pay attention to this point.
  • pls revise the Ref. carefully.

General comments:

  • The manuscript contains some typo errors; please revise it very carefully. A careful revision of the English grammar is required.
  • To summarize, this is a good study, which certainly merits a publication after a major revision.

Round 2

Reviewer 2 Report

Reviewer comments-R2

Agronomy (ISSN 2073-4395)

Manuscript ID: agronomy-1519779

Manuscript Title: Soil and plant responses to phosphorus inputs from different phytase-associated animal diets"

Reviewer comments:

In this study, the authors studied the effects of “Soil and plant responses to phosphorus inputs from different phytase-associated animal diets”

  • After reviewing the whole parts of MS and after reviewing the author’s response to my comments, I can say that this article is valid for publication in agronomy-basil after making some modifications to the English language. To summarize, this is a good study, which certainly merits a publication after a minor revision.